# Smartphone Application for Structural Health Monitoring of Bridges

**DOI:** 10.3390/s22218483

**Published:** 2022-11-04

**Authors:** Eloi Figueiredo, Ionut Moldovan, Pedro Alves, Hugo Rebelo, Laura Souza

**Affiliations:** 1Faculty of Engineering, Lusófona University, Campo Grande 376, 1749-024 Lisboa, Portugal; 2CERIS, Instituto Superior Técnico, University of Lisbon, Rovisco Pais, 1049-001 Lisboa, Portugal; 3COPELABS, Computer Engineering Department, Lusófona University, Campo Grande 376, 1749-024 Lisboa, Portugal; 4Applied Electromagnetism Laboratory, Universidade Federal do Pará, R. Augusto Corrêa, Guamá 01, Belém 66053-260, Brazil

**Keywords:** structural health monitoring, smartphone application, damage identification, machine learning, structural dynamics

## Abstract

The broad availability and low cost of smartphones have justified their use for structural health monitoring (SHM) of bridges. This paper presents a smartphone application called App4SHM, as a customized SHM process for damage detection. App4SHM interrogates the phone’s internal accelerometer to measure accelerations, estimates the natural frequencies, and compares them with a reference data set through a machine learning algorithm properly trained to detect damage in almost real time. The application is tested on data sets from a laboratory beam structure and two twin post-tensioned concrete bridges. The results show that App4SHM retrieves the natural frequencies with reliable precision and performs accurate damage detection, promising to be a low-cost solution for long-term SHM. It can also be used in the context of scheduled bridge inspections or to assess bridges’ condition after catastrophic events.

## 1. Introduction

Investment in road and rail infrastructure has been extensive in most developed countries, and bridges have proved to be among the most vulnerable parts of these assets, as observed by the number of collapses and their dire consequences.

Bridge owners use bridge management systems (BMS) to ensure a coordinated approach for optimizing costs, risks, serviceability, and sustainability of bridges. Initially, BMSs were simply inventories of basic information about bridges, such as construction date, location, and some structural details [1]. Then, they evolved to incorporate information derived from scheduled inspections and maintenance activities. Currently, BMSs intend to cover all activities performed during the service life of bridges, from design to demolition, taking into account public safety, budgetary constraints, and transport network functionality. They possess mechanisms to ensure that bridges are regularly inspected, evaluated, and maintained in a systematic way [2]. Broadly speaking, a BMS aims to ensure safety while minimizing costs. However, BMSs still rely heavily on bridge inspections, especially on the qualitative, and not necessarily consistent, visual inspections. This shortcoming may compromise the structural evaluation, the maintenance decisions, and ultimately the prevention of bridge failures [1].

Following the increased awareness of the limitations of visual inspection, in the last three decades, the non-destructive evaluation (NDE) and the structural health monitoring (SHM) fields have emerged to add more quantitative information into BMSs. NDE concerns the health assessment of a structure, or its components, through offline non-damaging procedures [3]. On the other hand, SHM assumes an online approach, continuous in time and global in nature, with the aim of autonomous monitoring. It should be noted that SHM is more than just monitoring, as simply collecting data does not constitute SHM. Rather, SHM assumes a continuous strategy of damage identification based on monitoring and interpreting the collected data. Therefore, it is fair to observe that NDE may be incorporated into SHM systems, but not vice-versa [1].

In order to facilitate the automatic interpretation of monitoring data and damage identification, SHM has been posed in the context of a statistical pattern recognition (SPR), which is traditionally split into four stages: (i) operational evaluation, (ii) data acquisition, (iii) feature extraction and generation, and (iv) statistical modeling for feature classification. This paradigm has been rooted in the artificial intelligence field, where machine learning algorithms play an essential role for pattern recognition [4]. Damage identification has been defined as a five-level hierarchy [5]: detection, localization, type, severity, and prognosis.

Despite the potential benefits of SHM for the authorities, it is widely recognized that bridge SHM is still not fully capable of producing reliable information on the presence of damage in a full-field monitoring strategy [6]. According to Cawley [7], in order to speed up the knowledge transfer from research to practical application of SHM in bridges, better-focused research and development is needed. Therefore, the scientific community has tried to find out new ways to improve the damage identification process. Recent technologies have unveiled alternative perspectives to manage and observe the response of monitored structures. In particular, we have observed recent breakthroughs in digital advancements, namely in (1) mobile device-assisted monitoring, (2) unmanned aerial vehicles, (3) virtual and augmented realities, and (4) digital twins [8].

In line with the general digitalization trend, the broad availability and low cost of smartphones have justified the recent investment in their use for SHM of bridges. Smartphones contain built-in sensors, storage capabilities, processors, and broadband communication networks, such as the cellular 4/5G network, Wi-Fi, and Bluetooth. Built-in sensors include accelerometers, cameras, microphones, gyroscopes, magnetometers, Global Positioning System (GPS) systems, LiDAR sensors, etc. A summary of the smartphone-related sensor technology and its application in SHM is made in reference [8]. This type of technology is relevant, as it can be used either as SHM systems or as NDE during bridge inspections.

Two main sensing strategies have been pursued, namely the use of a smartphone’s camera for displacement, deformation, and crack measurements through digital image correlation [9,10], and the use of a smartphone’s accelerometer to measure accelerations and estimate natural frequencies of vibration [11]. Even though the use of cameras has gained significant traction in SHM [12], especially in the last decade, this paper is especially concerned with the use of built-in accelerometers.

Since 2015, several publications have reported the applicability of smartphones for SHM of bridges using vibrations measured from the internal accelerometers. Zhao et al. [13] showed the applicability of smartphones to measure internal forces of cables, both in laboratory conditions and on the Xinghai Bay Bridge in China. In 2016, Zhao et al. [14] carried out a comprehensive experimental study on the deck and cables of a laboratory-scale suspended bridge to verify the feasibility of smartphones for a rapid damage assessment of bridges after an extreme event, such as an earthquake. The vibration responses obtained from smartphone accelerometers and conventional sensors were compared in both time and frequency domains. The measurements from smartphones agreed well with the measurements from reference accelerometers in terms of amplitude characteristics. The fundamental natural frequency of the bridge model was estimated with a maximum error of 0.44%.

Feng et al. [15] performed a comprehensive experimental study, involving seismic shaking-table tests and bridge field tests. Three standard smartphones with different accelerometers were compared with a high-quality reference/conventional sensor. The shaking-table tests confirmed that the smartphone sensors were capable of accurately measuring sinusoidal vibration between 0.5 Hz and 20 Hz. The authors also verified that the accelerometers embedded in newer generation smartphones are significantly more accurate than those in the older generations, which suggests that accelerometers are quickly evolving over time. The laboratory and field tests also showed the advantages of smartphone sensors over conventional sensors, such as the ease of installation and data acquisition, as well as wireless transmission.

Using two smartphones in the laboratory, Ozer et al. [11] showed how response measurements from cameras and accelerometers can be blended for experimental modal analysis, benefitting from the accessibility of smartphone technology to provide innovative SHM solutions with mobile, programmable, and low-cost interfaces.

Mei and Gul [16] proposed a crowdsourcing-based methodology for monitoring and evaluation of a population of bridges using the accelerometers mounted in smartphones in a large number of moving vehicles as mobile sensors. The authors showed that the methodology could provide damage features correlated with the severity of damage. Other authors have shown in laboratory experiments the feasibility of bridge frequency estimation using multi-vehicle (i.e., crowdsourced) smartphone accelerations [17].

Shrestha et al. [18] implemented a smartphone-based monitoring system for more than a year covering one bridge in Japan. Six smartphones were streaming accelerations to a cloud server to verify their viability and accuracy. Dynamic properties estimated from this system were compared with those of reference seismometers. Inconsistent sampling rates and faulty sensor readings were identified to be two major problems with the use of the proposed system.

Recently, Ozer et al. [19] used multiple smartphones throughout the Golden Gate Bridge in the United States. The accelerometer data were collected to retrieve modal frequencies and mode shapes of the bridge. Despite the low vibration frequency and amplitude of a long-span suspension bridge combined with limited sensing and acquisition quality of the smartphones, the authors found significant correlation with the modal properties estimated with conventional sensing technology.

In conclusion, early studies assessed the applicability, precision, and resolution of smartphone vibration measurements and concluded that they are reliable in the frequency range of interest for bridges (<20 Hz).

Previous attempts to harvest the capability of the smartphones to recognize and measure frequencies that lay in the effective frequency range have been reported. An effort which is relatively similar to App4SHM was reported by Ozer et al. in 2015 [20]. The authors created CS4SHM, an application that crowdsourced smartphone measurements on flexible bridges and extracted the frequency with the largest spectral coefficient from the acceleration time series. Besides the fact that CS4SHM was developed under IoS, the main difference in respect to App4SHM is that it included no damage detection component, although the authors plan to add that in future developments. Another excellent effort is Orion-CC, an IoS smartphone application for cable force measurement based on monitoring the vibration of the cable [13].

This paper proposes a novel smartphone application for damage detection in bridges under ambient vibration conditions, making use of the built-in accelerometer and following the SPR paradigm. The application is not limited to cables, nor to frequency extraction, as was the case of the applications cited above. Instead, it includes a machine learning algorithm capable of recognizing damage after being trained with data from the undamaged structure. The application is first tested on data sets from a laboratory steel beam to verify the sensing and damage detection capabilities and then it is tested on two twin post-tensioned concrete bridges for damage detection. In both cases, the natural frequencies obtained from the application are compared with those estimated using data sets from a conventional data acquisition system.

App4SHM is particularly well-suited to periodically monitor bridges under operational and environmental conditions. It was developed in a joint effort between the Civil Research Group and the Computer Science Department from the Lusófona University in Portugal. It can be applied to perform SHM or NDE during scheduled inspections, or to assess the structural condition after a catastrophic event, or when required by authorities and stakeholders. It may also be regarded as NDE technology integrated into the SHM of bridges. In the damage identification hierarchy, this application only addresses the first level, related to damage detection.

App4SHM is an Android application already available in the Google Play Store to a limited group of testers, selected upon request after the application was presented at the 10th European Workshop on Structural Health Monitoring [21]. We are planning to make it generally available as soon as the testing phase ends. App4SHM fits into a larger effort to transpose SHM technology from research to practice by developing innovative, low cost data acquisition systems along with supporting software (e.g., [22,23]).

The organization of this article is as follows: besides this section, Section 2 describes the basic architecture of the proposed smartphone application. Section 3 presents and discusses results from laboratory tests conducted on a simply supported steel beam. Section 4 presents the results obtained from ambient vibration tests carried out on two twin concrete bridges. Finally, Section 5 provides the main conclusions of this paper and gives some indications of future works.

## 2. Software Architecture and User Interfaces

### 2.1. Architecture Description

App4SHM (Figure 1) is an application that interrogates the phone’s internal accelerometer to measure accelerations, estimate the natural frequencies, and compare them with a reference data set through a machine learning algorithm properly trained to detect damage in almost real time. By ‘almost real time’, we understand the time needed for data acquisition, feature extraction, and damage detection, typically taking less than two minutes altogether. App4SHM includes access to a server to run most of the computational operations and store some of the data sets (see Figure 2).

The application is composed of two software components: a mobile application and a computational platform hosted by the server. The mobile application is developed in Kotlin and is compatible with devices running Android 10 operating system (or higher). It implements a navigation mechanism with four tabs (represented by the green boxes in Figure 2), corresponding to the four steps in the damage detection process:Structure Identification (Step 1)—identification of the structure from a structural database. A new structure can be added via server;Data Acquisition (Step 2)—measurement of acceleration time series in the direction orthogonal to the screen of the smartphone (Figure 1) for a user-controlled period of time;Feature Extraction (Step 3)—estimation of the first three natural frequencies from the power spectral density of the accelerogram (using the Welch method). They are stored into a feature vector (observation);Damage Detection (Step 4)—for each new observation, a damage indicator is computed based on the Mahalanobis-squared distance (as applied in [24,25]). Subsequently, a plot of the damage indicators of the observations collected in the training database (and assumed to reflect the behavior of the undamaged structure) is presented, to which the damage indicator computed for the new observation is added in green or red if the structure is deemed undamaged or damaged, respectively.

The application is linked with a Python/Django server through REST webservices to get existing structures, perform calculations, and save the measured structural frequencies in a MySQL database (orange boxes in Figure 2). There is also a web interface (Figure 3), accessible with any browser, providing a backoffice (automatically generated by Django admin) which allows users to manage all the information stored in the database: structures, time series, power spectral densities, and natural frequencies. Using this interface, users with administrator’s privileges may access all monitoring information, including accelerograms, power spectral densities, and extracted frequencies. It is also possible to upload previously recorded observations, register new structures, and delete incorrect data sets, among other operations. This architecture has the following benefits:Multiple smartphones can be used to record acceleration time series for the same structure and centralize the information in a single database;For structures with existing training sets, it is possible to upload that information directly to the database and use it along data coming from the smartphone measurements to train the machine learning algorithms for damage detection;The web-based backoffice is accessible only to users with the right credentials, meaning that regular users cannot tamper with the information stored in the database;CPU-intensive calculations such as the Welch algorithm can be slow on certain mobile devices, so it is useful to offload those computations to the server;Facilitates the development of the application on other platforms (namely iOS) since the backoffice algorithms implemented on the server need not be changed.

### 2.2. User Interfaces

The user interfaces guiding the interaction between user and smartphone (Figure 4) are described below in more details.

#### 2.2.1. Step 1—Structure Identification

This step starts by calling a webservice to get the identification of the structures for which saved frequencies already exist in the database (Figure 4a). This procedure allows the user to choose an existing structure (the amount of stored training observations is presented next to the name of each structure). It also defines the protocol for the collection of a new observation: either for adding it to the training database or for damage detection purposes. If the goal is training, the process ends with the feature extraction step and no damage indicator is computed. Conversely, under the damage detection protocol, a damage indicator is computed, but the observation is not stored in the training database. As mentioned above, the creation of a new structure is only possible via the server.

#### 2.2.2. Step 2—Data Acquisition

In this step, App4SHM collects information from the built-in accelerometer of the smartphone with a sampling period of approximately 20 ms (50 Hz). As smartphones have imperfect sampling frequencies [16], it is not possible to guarantee a precise sampling rate. Therefore, an interpolation algorithm is implemented to correct deviations from the defined sampling frequency. As smartphones are usually laid down over the structure, only accelerometer data in the z axis (see Figure 1) is currently considered. Collected accelerations are drawn in a graph in real-time to assist the user during this step (Figure 4b). They are also locally stored in the smartphone in CSV format (a format that is recognized by most spreadsheet applications).

#### 2.2.3. Step 3—Feature (Natural Frequencies) Extraction

The mobile application sends the acceleration data to the server, using a webservice, to extract frequencies. The frequency extraction process is comprised of two parts. First, a power spectral density analysis is done in the server using the Welch algorithm. For this purpose, the measurement interval is divided into a number of overlapping intervals. Both the number of intervals (default = 3) and the overlapping time (default = 50% of the interval) can be controlled programmatically. The power spectral density returned to the user is the average of the spectra for each of the intervals. Then, the power spectral density plot is sent back to the smartphone and shown in the mobile application (see Figure 4c). The spectral density information is stored on the smartphone. The user selects three frequencies which are then sent to the server to be saved in the database X (in case of training) or to serve as input to the damage detection algorithm. To assist the user with the selection, the average values of the training frequencies already stored on the server are presented. Unlike alternative applications like CS4SHM [20], App4SHM purposefully does not automatize the frequency extraction process from the power spectral density. This option is justified by our policy of keeping App4SHM a professional (rather than a crowdsourcing) tool. It is noted that, as opposed to CS4SHM, App4SHM has a damage detection module which may induce concern to untrained users, especially if the frequency data measured on the undamaged structure is not sufficient to be statistically relevant.

#### 2.2.4. Step 4—Damage Detection

The damage detection step applies a Mahalanobis-based algorithm to detect abnormal observations by measuring the distance between the previously recorded frequencies and the current observation. This results in a set of damage indicators (*DI*), which is transmitted to and shown in the mobile application (Figure 4d). If the damage indicator is superior to a threshold defined for a level of significance of 5% of a chi-squared distribution, the application raises a red flag and outputs a warning message. Otherwise, App4SHM raises a green flag, as no damage indication is present. The 5% level of significance is a common choice in the literature (e.g., [26]), as it strikes a good balance between the risks of the algorithm yielding false positives and false negatives. However, the level of significance can be easily changed programmatically, if needed.

The choice of the Mahalanobis algorithm for damage detection is grounded in its simplicity and generalization capabilities. Its effectiveness is demonstrated in multiple papers (e.g., [25,27]) that apply it to data sets from several real-world bridges, including comparisons with alternative algorithms. The assumption of normally distributed data allows the definition of thresholds for given levels of significance. However, other algorithms can be added programmatically to App4SHM if desired, by substituting a single function on the supporting server.

As proposed by Figueiredo et al. [24,25] in the context of bridges, considering the training matrix $X\in {ℜ}^{n\times m}$, where n is the number of observations and m is the number of variables, with multivariate mean vector, μ, and covariance matrix, Σ, the Mahalanobis squared distance (or DI in the context of this study), between feature vectors of the training matrix and any new feature vector (or observation) is defined as:(1)$$DI={\left(z-\mu \right)}^{T}\Sigma^{-1}(z-\mu )$$

The assumption is that if a new observation **z** is obtained from data collected on the damaged system, the observation will be further away from the mean of the normal condition. On the other hand, if an observation is obtained from a system within its undamaged condition, even with operational and environmental variability, it will be closer to the mean of the normal condition.

Herein, for feature classification, it is assumed that if an observation **z** is extracted from an undamaged condition and it is multi-dimensional Gaussian distributed, then its DI will be Chi-square distributed with *m* degrees of freedom, $DI\sim \chi_{m}^{2}$. Therefore, multivariate outliers can now simply be defined as observations having *DI*s above a certain level of confidence.

## 3. Case Study #1: Simply Supported Beam

Two key questions regarding the use of smartphones for SHM can be posed as follows: is the (poor) sensitivity of their accelerometers enough to endorse the correct calculation of the frequencies needed for damage detection? And how large should the damage be to be recognizable by the application? This section firstly assesses the quality of the smartphone’s measurements in terms of accelerations, power spectral density, and natural frequencies. Subsequently, the damage detection capabilities of App4SHM are evaluated.

A simply supported steel beam was used to compare both the time series and the power spectral densities obtained using App4SHM and a conventional data acquisition system.

### 3.1. Structural Description

The steel beam, illustrated in Figure 5, was composed of two equal leg angles (L 20 × 20 × 3 mm) connected at given intervals by 100 × 3 mm plates to ensure that a global flexural behavior is achieved. The steel beam was simply supported with a free span of 4480 mm.

### 3.2. Vibration Tests

The steel beam was subjected to non-destructive forced vibration tests using an impact hammer, a conventional sensing system composed of two piezoelectric uniaxial accelerometers from PCB Piezotronics, Inc. (Depew, NY, USA), and a smartphone. The two piezoelectric accelerometers, with a sensitivity of 10 and 0.5 V/g and a measurement range of ±0.5 and ±10 g, were fixed on the underside of the steel plate by means of an auxiliary base, which was glued to the steel structure using a quick bonding gel. Figure 6 depicts the two piezoelectric accelerometers used in the laboratory tests. Additionally, the piezoelectric accelerometers required the use of a portable PC, running LabViewTM rel 11 in Windows, a digital analogue card, and a signal conditioning module. Alternatively, a smartphone was simply placed on the topside of the same steel plate and required no additional equipment. A total of 30 non-destructive impact tests were performed, resulting in the corresponding 20 s time series from the conventional sensing system and smartphone. The (extremely short) impact time is removed from the time series that are analyzed for natural frequencies.

### 3.3. Comparative Study

The comparative study was first carried out in terms of the first three natural frequencies, which were computed resorting to a peak-picking method on a power spectral density determined using the Welch algorithm. The accuracy of the natural frequencies is instrumental for the functioning of the damage detection algorithms. Next, in order to conduct the remainder of the comparative study, two of the 30 non-destructive tests were chosen. These tests correspond to those that deviate the most and the least from the average in terms of natural frequencies. The time series and power spectral densities, determined using the two accelerometers and the smartphone, were compared both qualitatively and by means of time-lagged cross-correlation. The latter allows one to automatically synchronize the conventional sensing system and smartphone’s time series and determine the maximum cross-correlation between them. It is noted that, unlike the correct estimation of the natural frequencies, the good correlations between the power spectral densities and (even more so) between the acceleration time series are not instrumental for the damage detection algorithms. However, the analysis of the correlation between the measurements obtained using the conventional sensing system and the smartphone supports a more complete perspective of the limitations of the latter as compared to the former.

#### 3.3.1. Natural Frequencies

The first three natural frequencies for the 30 time series are estimated and presented in Table 1. Additionally, the mean, standard deviation, and confidence interval (95% confidence level) were determined for all tests. Comparing the average frequencies obtained resorting to the smartphone with those determined using the conventional sensing system, a maximum difference of 0.54% is obtained. Consequently, one may conclude that the smartphone was able to consistently determine the natural frequencies when compared with a conventional sensing system. Moreover, when analyzing the observations’ standard deviation, the smartphone presented a smaller standard deviation regardless of the considered natural frequency, which indicates excellent consistency.

Additionally, a correlation test was made to verify if the natural frequencies computed resorting to the conventional accelerometers and smartphone are linearly related to each other. Figure 7 depicts a matrix with the correlation heatmap, where the size and color intensity of the square are proportional to the correlation’s magnitude, while the color itself separates the negative and positive correlations. A strong positive correlation is attained between the conventional accelerometers, with correlation values between 0.84 and 1.00 obtained when comparing the same frequencies. The correlation values between the smartphone and the accelerometers are smaller (between 0.63 and 0.76), but still strongly positive. Lastly, if one compares different frequencies, regardless of the considered accelerometer, only small correlations were observed as suggested by the off-diagonal entry values.

Due to the strong positive correlation observed for the same frequencies, one may conclude that an excellent agreement between the natural frequencies computed using the smartphone and those determined using conventional sensing systems was attained.

The ANOVA test is commonly used to analyze the differences between the means of several populations based on their variances. It was used in the present study to verify the similarity between the average frequencies attained with different accelerometers. Through the comparison of the values attained for the first, second, and third natural frequencies, the ANOVA test determined *p*-values of 0.77, 0.99 and 0.68, respectively. Therefore, since a *p*-value larger than 0.05 indicates a valid null hypothesis (population means are equal), one may conclude that the frequencies computed using the reported sensing systems are statistically similar.

Although the main output of App4SHM is the indication of damage based on the first three natural frequencies of a given structure, a more in-depth analysis is made in terms of the recorded time series and power spectral densities. The following two subsections present the results.

#### 3.3.2. Time Series

Due to the impracticality of testing all the available time series, and due to their similarity in terms of natural frequencies, only two sets of time series were chosen for further analysis. These sets, which are highlighted in Table 1, are the closest and furthest away from the average values of the frequencies. Firstly, the time series obtained using the conventional sensing system were compared in order to verify their similarity, both qualitatively and by means of cross-correlation. Next, if the readings of the accelerometers are highly similar, only one of their time series is compared with the time series recorded using the smartphone.

Figure 8 illustrates the comparison between the time series obtained resorting to the two accelerometers of the conventional sensing system. As expected, an excellent agreement is observed between the two accelerometers for all tests. Alternatively, the cross-correlation method, which determines the similarity between a fixed time series and a time-lagged one as a function of the lag, was also used to compare the time series. As expected, a null time lag was achieved for both tests, while correlation values of 0.97 and 0.96 were attained, indicating a strong positive correlation between the acceleration time series.

Therefore, the time series recorded by the smartphone is only compared with one of the conventional sensing systems, namely the 333B40 accelerometer, due to its larger measurement range.

Figure 9 depicts the time series obtained using the smartphone and the 333B40 accelerometer. The signals have a good general agreement, presenting only a limited number of small differences. Resorting to the cross-correlation method in order to synchronize the time series, correlation values of 0.71 and 0.55 were achieved for tests number 1 and 8, respectively. These values are considerably lower than those obtained for the time series measured with conventional accelerometers. This is caused by the relatively poor sensitivity of the smartphone’s accelerometer as compared with professional equipment. However, despite the difference between the time series, their frequency content is remarkably similar, as shown in the next section.

#### 3.3.3. Power Spectral Densities

The entire set of time series obtained during the experimental study was used to compute their respective power spectra, which were determined using the Welch algorithm. Figure 10 illustrates all power spectral densities in grey lines, along with their averages, in blue lines, for the smartphone and conventional accelerometers. Similar to that reported in Table 1, the analysis of Figure 10 reveals a low variability in terms of natural frequencies, i.e., the local maxima, regardless of the considered accelerometer.

Additionally, Figure 11 depicts the average power spectral densities (using a logarithmic scale) obtained with the three devices, which exhibit an excellent agreement concerning the structure’s natural frequencies. Moreover, in order to compare the entire frequency range of the power spectral density, a cross correlation analysis was performed. This analysis yielded correlation values of 1.00 and 0.99 between each accelerometer and the smartphone, which reveals an excellent agreement between the considered power spectral densities.

### 3.4. Damage Detection Capabilites

An additional experimental campaign was performed to evaluate the damage detection capabilities of App4SHM. One hundred dynamic impulse tests were conducted on the simply supported steel beam presented in Figure 5 to extract its first three natural frequencies under normal (‘undamaged’) conditions. These observations yield the standardized natural frequencies cluster illustrated with blue circles in Figure 12. The standardization was performed by subtracting the mean of the training data from each frequency and dividing the result by the standard deviation.

Subsequently, three different masses (25, 50 and 100 g, see Figure 13) were placed at two different locations, namely at mid-span (position A in Figure 5) and at one quarter of the span (position B in Figure 5), to simulate several levels of damage and, consequently, identify the damage required by App4SHM to yield a positive reading. It is noted that the total mass of the structure is 9.7 kg. The addition of mass to simulate damage is a relatively standard procedure when the structure cannot be damaged, e.g., [28]. 

Five dynamic tests were executed for each mass/location combination and the first three natural frequencies were extracted for each case, yielding the squares, diamonds and upper triangles illustrated in Figure 12. Note that the marker indicates the amount of mass added to the steel beam, whilst the marker’s color specifies if damage was identified by App4SHM (red) or not (green). Figure 12a, which illustrates the results obtained when the mass is placed at mid-span, shows that the first and (especially) third frequencies are the most affected, while the second frequency suffered no significant change. This is physically consistent, as the second natural mode of vibration has a node at mid-span, so should be less affected by an additional mass placed at that position. Figure 12 shows that App4SHM correctly identifies damage when at least one natural frequency deviates by more than two standard deviations from the training cluster.

The clusters obtained when the mass is located at approximately one quarter of the span are presented in Figure 12b. In this case, the second frequency was by far the most affected, with limited degradation of the first and third frequencies. Again, the insensitivity observed for the third frequency is to be expected as the added masses were located close to a vibration node of the third modal shape. Conversely, the location corresponds to an anti-node of the second mode of vibration, which is therefore the most affected. Similarly to the observations made when the mass was placed at mid-span, App4SHM starts detecting damage when the differences between the normal and measured frequencies exceed two standard deviations in at least one natural frequency.

The evolution of the damage index (*DI*), defined by Equation (1), as a function of the added mass and its location is presented in Figure 14. Each cyan circular marker corresponds to a damage detection observation, the circle with a dot represents the mean *DI*, and the vertical lines indicate the ± standard deviation range. The *DI* threshold that separates undamaged from damaged data is shown as a horizontal line. The lowest added mass tends to yield higher *DI*s than the training data, but they always remain below the threshold. The 50 g mass is recognized as damage in most cases, especially when the mass is located at mid-span. However, the threshold is located at less than one standard deviation from the mean, so it is expected that the Mahalanobis-based algorithm would yield a considerable number of false negatives. Finally, the highest mass has a marked impact on the *DI*, with the measurements always located more than two standard deviations higher than the threshold. This means that false negatives should very seldomly occur for this level of added mass. Lastly, as expected, placing the mass at mid-span leads to larger *DI*s as compared to those attained with the mass at one quarter of the span.

For comparison purposes, the damage detection analysis was also performed using an alternative algorithm, based on Gaussian Mixture Models with an adaptive choice of the number of clusters to define the baseline condition [24]. The algorithms yield the same classification for the readings, as listed in Table 2.

## 4. Case Study #2: Twin Bridges over Itacaiúnas River

### 4.1. Structural Description

Two side by side “twin” bridges (Figure 15) over the Itacaiúnas River in Brazil and integrated in the Transamazônica Road were used to test the App4SHM under real-world conditions. One of the bridges (old bridge) was built in the 1980s and the other (new bridge) was inaugured in 2010 due to the increase of traffic. Both bridges are post-tensioned concrete box-girder bridges with a main span of 120 m and two side spans of 70 m each. Both intermediate supports are concrete piers resting on concrete piles and clamped into the girder. Small structural differences exist between the bridges, most notably the height of the piers (from the girder to the cap of the piles) being roughly 1 m larger on the old bridge. As was the case for the lab beam presented in the previous section, the tests were designed to compare the readings obtained with conventional accelerometers and App4SHM and to test the damage detection feature offered by the latter.

### 4.2. Ambient Vibration Tests

Ambient vibration tests were carried out in October 2021 on the two bridges to estimate the operational mode shapes, using a conventional sensing system composed of three-axis strong motion force balance accelerometers from GeoSIG. In previous tests carried out in 2018, nine checkpoints were used as shown in Figure 16. This time, additional checkpoints were added to increase the resolution of the mode shapes’ configuration, resulting in a total of 19 checkpoints. Checkpoints between sections 4 and 6 (see Figure 16) were designed to find possible cracks in that area. The checkpoint placed at section 4 was kept as a reference point throughout the ambient vibration tests. Each checkpoint is composed of one fixed accelerometer (reference) and two moving accelerometers.

The modal properties (natural frequencies, mode shapes, and damping ratios) were estimated through ARTeMIS Modal using the enhanced frequency domain decomposition (EFDD). Table 3 shows the operational mode shapes and natural frequencies for both bridges through two ambient vibration tests. The natural frequencies correspond to the average of the 19 checkpoints. As those two bridges are similar, the mode numbering is based on the direct comparison of the mode shapes. For the new bridge, the mode shape with natural frequency of 1.416 Hz is left behind, as there is no direct correspondence with the old bridge. Table 4 summarizes the estimated natural frequencies. In general, assuming both bridges have similar mass, the new bridge is more flexible than the old one.

### 4.3. Feature Extration and Comparative Study

A comparative study was conducted resorting to measurements in section 2 (see Figure 16) of the new bridge using both sensing systems. As the smartphone measurements are obtained from a section where frequencies F2, F3, and F4 are best read, these frequencies are used for comparison. The smartphone was placed directly on the sidewalk as shown in Figure 17.

As reference, three time series with 15 min duration each were obtained using the traditional sensing system (GeoSIG). For later comparison, three natural frequencies were estimated and are summarized in Table 5. The difference between the mean and the measured values is also summarized (see the values in the brackets). A maximum variation of 1.08% around the mean was observed in the estimated frequencies over one hour of testing, roughly.

The natural frequencies estimated using App4SHM are based on six time series of 20 s duration each obtained in the same one-hour time window (Table 6). The acquisition time was limited to ensure efficient communication between the mobile device and the server, under the rather modest 3G wireless mobile network available on the site. According to Table 6, the maximum variation of the App4SHM measurements as compared to the GeoSIG mean values (taken from Table 5) was −2.53%. The absolute values of the differences in the mean values are smaller, namely between 0.55% and 1.43%.

Comparing the means of both sensing systems and taking GeoSIG as reference, one can conclude that the App4SHM outputs consistent results in terms of natural frequencies (differences less than 1.43%, as shown in bold in Table 6), even using a short time-series duration.

Another study was carried out on the old bridge to study the variability of the measurements throughout the day. Twenty-five observations were made with the App4SHM at section 5, using the same amount of time series, during the ambient vibration test performed on the 15th of October 2021 using the traditional sensing system (GeoSIG). Table 7 summarizes the results from both sensing systems, which shows a maximum variation of 2.42% compared to the values obtained from the 6-h ambient vibration test (GeoSIG) at 19 checkpoints.

### 4.4. Damage Detection Performance

Finally, to test the damage detection performance of the App4SHM for long-term monitoring, 165 observations obtained at section 5 from the *new bridge* over the course of one year (120 measurements in October, 2021 and 45 weekly observations ever since) were included in the training data set of the Mahalanobis-based machine learning algorithm. Each observation is composed of three natural frequencies (F1, F3, and F5, see the corresponding modes in Table 3).

After the training step, two tests were performed independently on both bridges, using one observation from *each bridge* at section 5 and the training database from the new bridge only. The markers in Figure 18 show the *DI* for each observation obtained from the new and old bridges against the threshold defined for a level of significance of 5%. As expected, the results show that the observation from the old bridge is not statistically similar to the training data from the new bridge, giving an indication of some anomaly and recommending further analysis to discover the source of such outlier (the observation in red is highlighted inside a red circle, in Figure 18b). Conversely, the observation from the new bridge ranks below the threshold and no damage warning is issued (the observation in green is highlighted inside a green circle in Figure 18a).

## 5. Conclusions

Smartphone-based applications for SHM still face technical challenges because of operator-induced uncertainties during data acquisition and feature extraction, a lack of control over the embedded sensing technology, and relatively low sensitivity and resolution of their built-in sensors. However, the end-user technology provided by smartphones has the potential to transform the way we perform bridge inspections, especially after catastrophic events such as earthquakes and tornados.

As a customized SHM process for damage detection, App4SHM does not intend to replace the visual inspections, rather it intends to provide more accurate assessment during bridge inspections and, at most, to reduce their frequency.

App4SHM was tested on data sets from a laboratory steel beam and two twin real-world concrete bridges. The former was designed to understand its sensing and damage detection capabilities in a more controlled environment, and the latter for damage detection in real world scenarios.

From the laboratory experiments, a high correlation between time series and, particularly, power spectral densities from different sensing systems (smartphone and two conventional accelerometers) was found. Comparing the average frequencies obtained resorting to the smartphone with those determined using the conventional sensing system, a maximum difference of 0.54% was obtained. The lab beam was also used to assess the sensitivity of App4SHM to damage, simulated through the addition of three different weights at two locations on the structure. It was concluded that damage becomes detectable when at least one of the monitored frequencies differs in more than two standard variations from its mean value under undamaged conditions.

The analyses conducted on two twin concrete bridges show that in terms of feature extraction, App4SHM outputs natural frequencies that are consistent with those obtained using conventional accelerometers (differences < 2.42%), even using short time-series durations, and within the variability observed with the traditional sensing system. Moreover, App4SHM correctly identifies the existence of outliers when observations from one of the bridges are fed into machine learning algorithms trained with data from the other (structurally similar) bridge.

The results from the two case studies show that App4SHM is reliable for both vibration measurement and damage detection in the context of SHM. However, as is the case with all individual components of the SHM, App4SHM is not meant to be used unaccompanied, but integrated into a Bridge Management System, along with complementary sensors and NDE.

The app and the sensing technology of smartphones still have some limitations. However, the laboratory and field tests have shown certain applications where we can obtain a reasonable trade-off between the potential lack of sensitivity and the advantages provided by the ease of installation and data acquisition as well as wireless transmission.

## Figures and Tables

**Figure 1 sensors-22-08483-f001:**
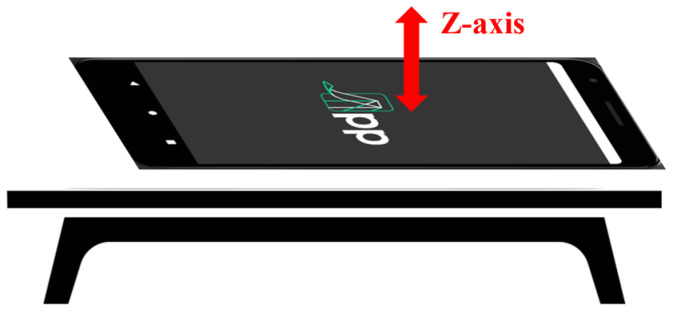
Measurement axis used in the application.

**Figure 2 sensors-22-08483-f002:**
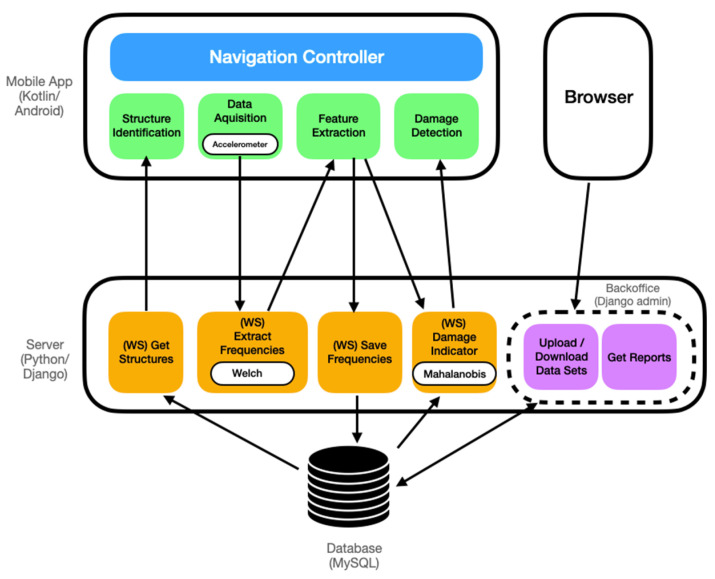
Architecture of App4SHM.

**Figure 3 sensors-22-08483-f003:**
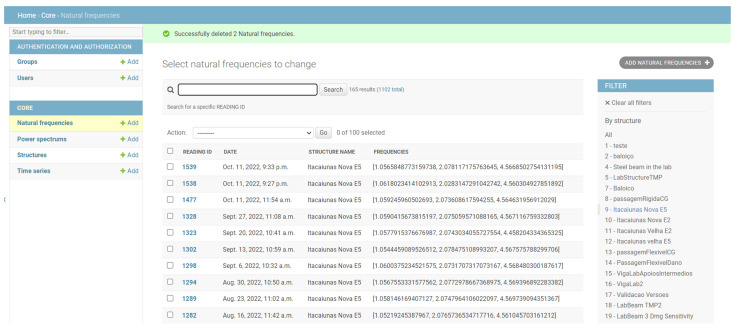
Backoffice administration of App4SHM.

**Figure 4 sensors-22-08483-f004:**
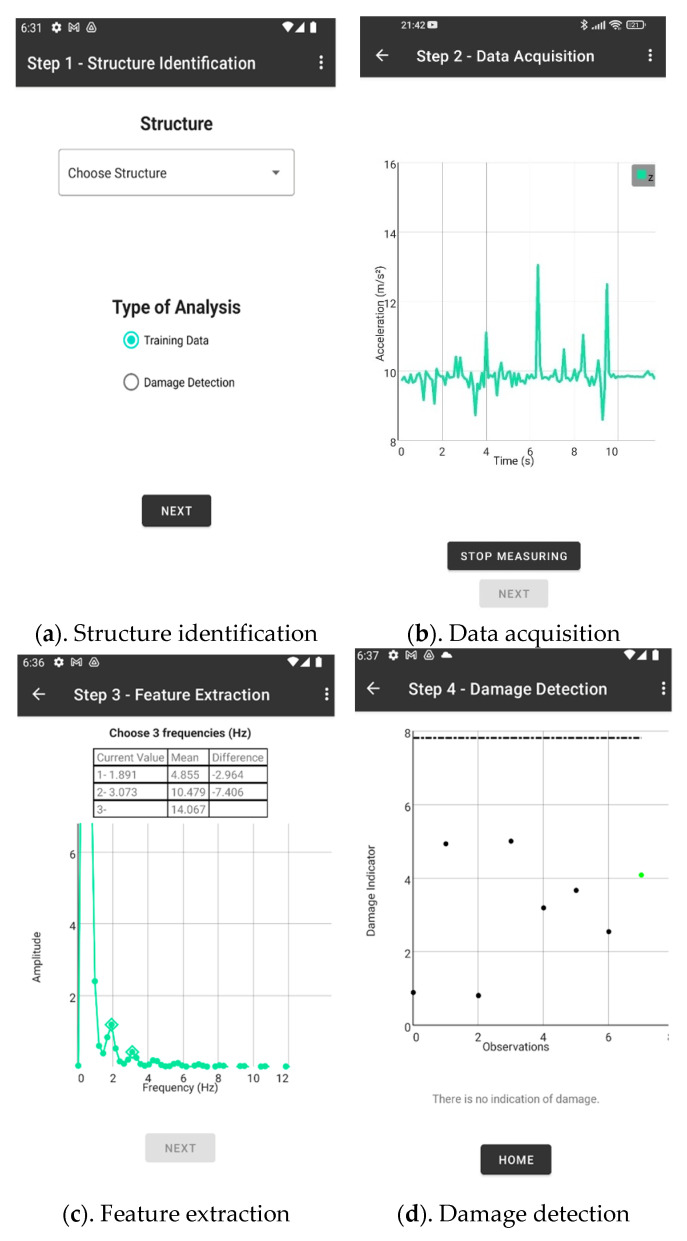
User interfaces of the application.

**Figure 5 sensors-22-08483-f005:**
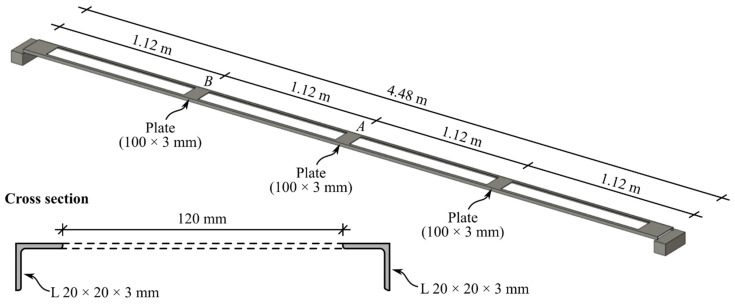
Schematic longitudinal representation of the steel beam along with typical cross section.

**Figure 6 sensors-22-08483-f006:**
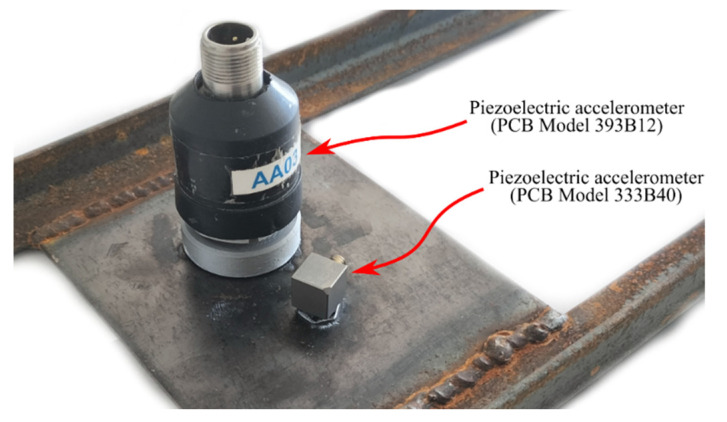
Two piezoelectric accelerometers fixed on the underside of the steel beam.

**Figure 7 sensors-22-08483-f007:**
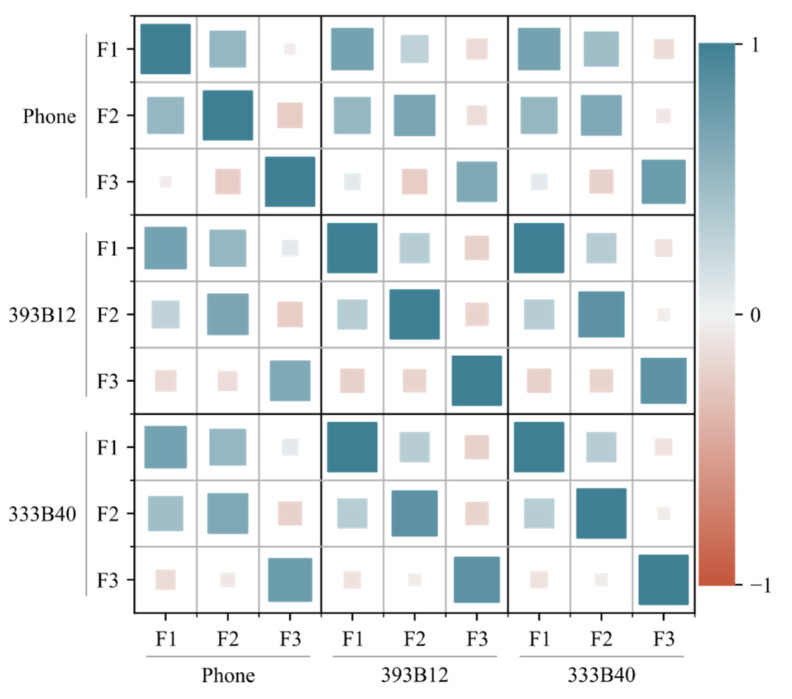
Correlation matrix of the computed natural frequencies.

**Figure 8 sensors-22-08483-f008:**
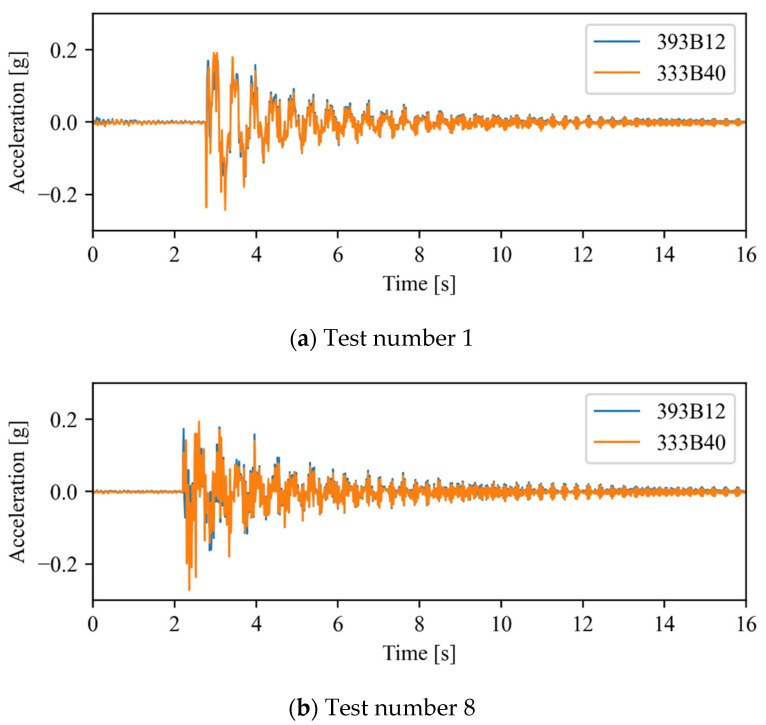
Time series comparison (393B12 vs. 333B40).

**Figure 9 sensors-22-08483-f009:**
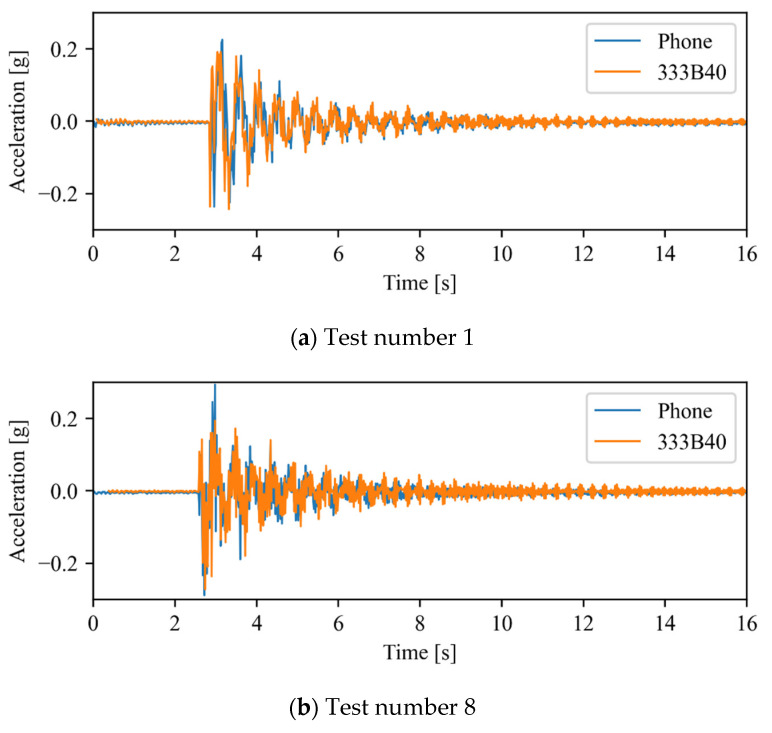
Time series comparison (Phone vs. 333B40).

**Figure 10 sensors-22-08483-f010:**
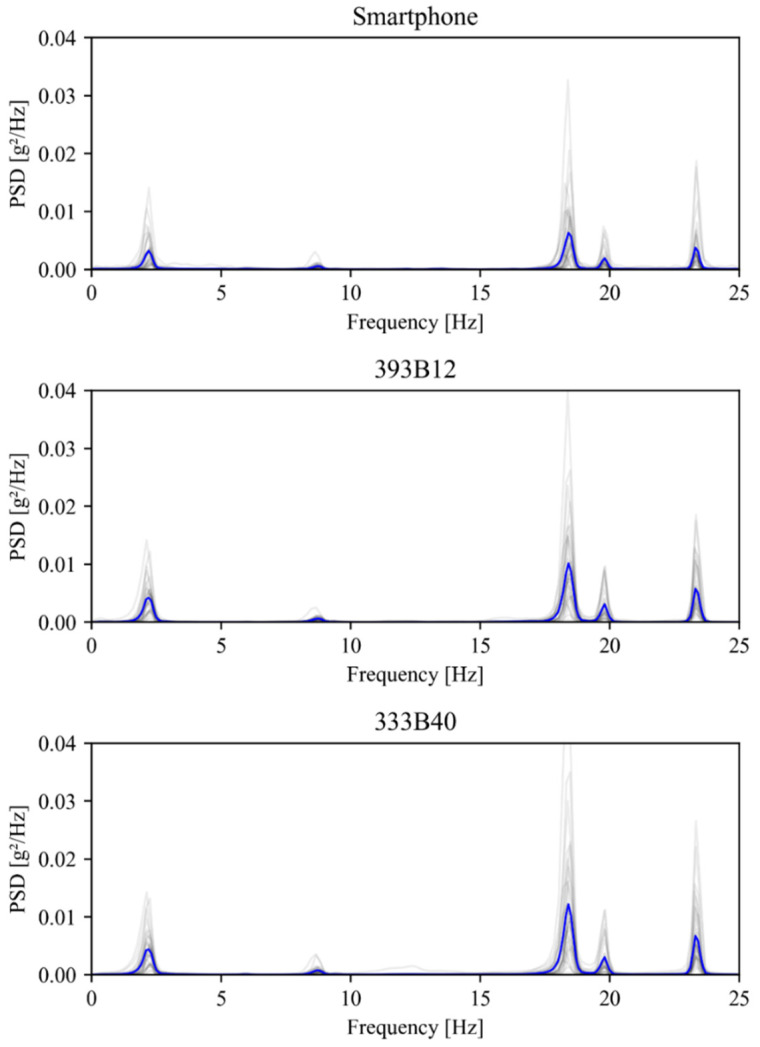
Individual (grey lines) and average (blue lines) power spectral densities (PSDs).

**Figure 11 sensors-22-08483-f011:**
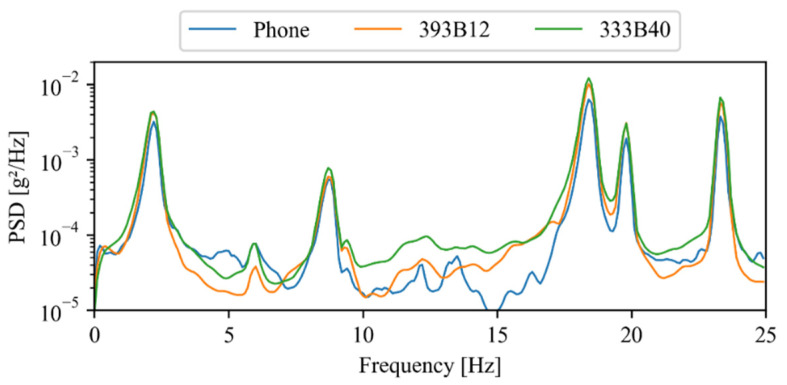
Average power spectral densities.

**Figure 12 sensors-22-08483-f012:**
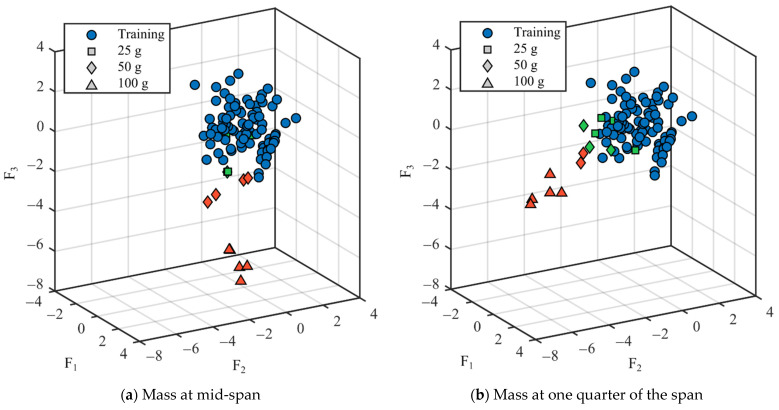
Standardized natural frequencies. The training database is represented with blue circles. The points corresponding to the three levels of damage are plotted in green if they were classified as ‘undamaged’ and in red if they were classified as ‘damaged’.

**Figure 13 sensors-22-08483-f013:**
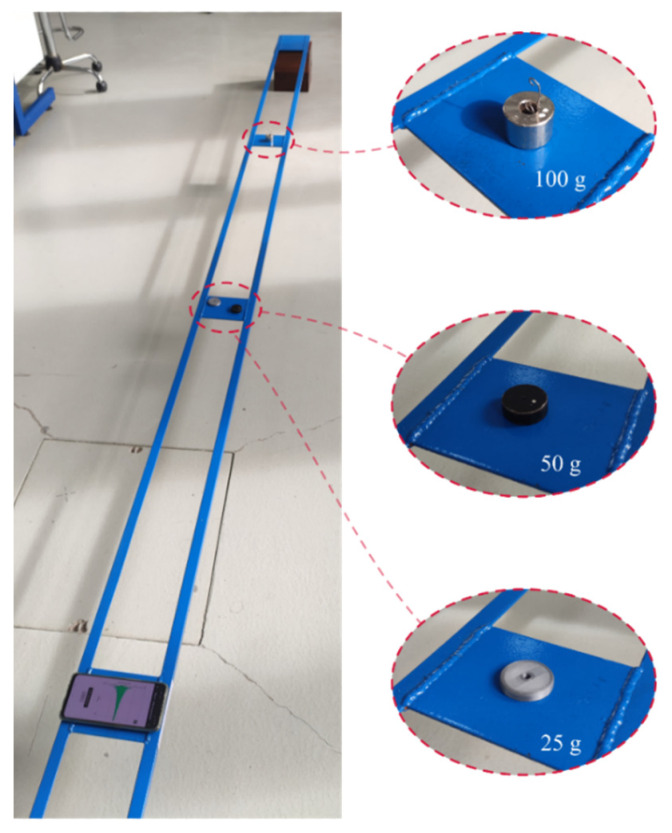
Lab beam and added masses used to simulate damage.

**Figure 14 sensors-22-08483-f014:**
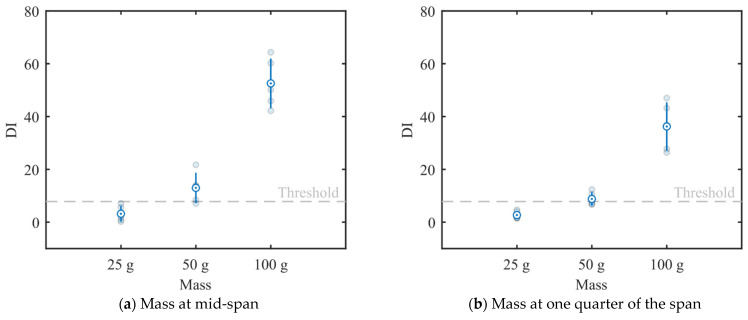
Damage index as a function of the added mass. Circular markers represent the five observations made for each added mass. The mean and standard deviation of each dataset are presented as the blue circle with a dot and the vertical blue bars, respectively.

**Figure 15 sensors-22-08483-f015:**
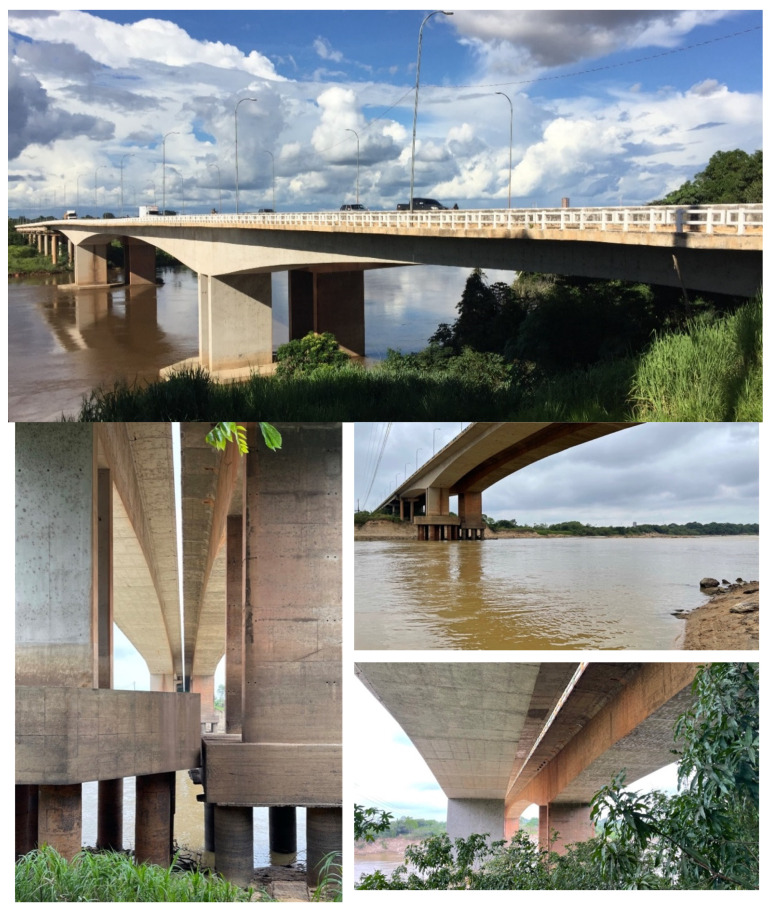
New (**left**) and old (**right**) bridges over the Itacaiúnas River.

**Figure 16 sensors-22-08483-f016:**
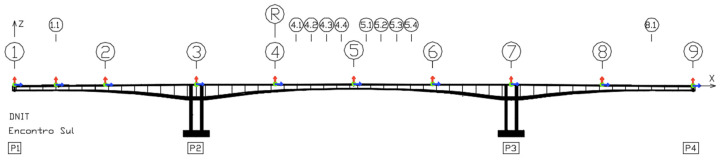
Configuration of the measurement sections used to perform ambient vibration tests.

**Figure 17 sensors-22-08483-f017:**
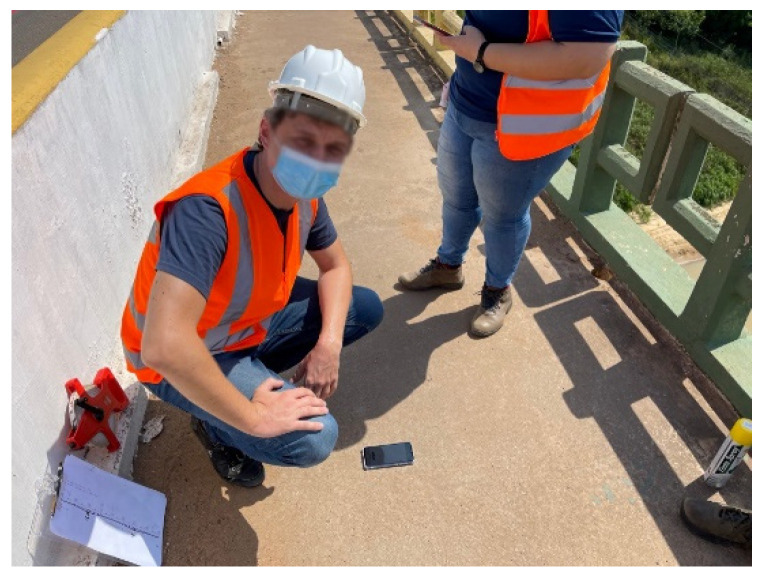
App4SHM being tested at the Itacaiúnas bridges.

**Figure 18 sensors-22-08483-f018:**
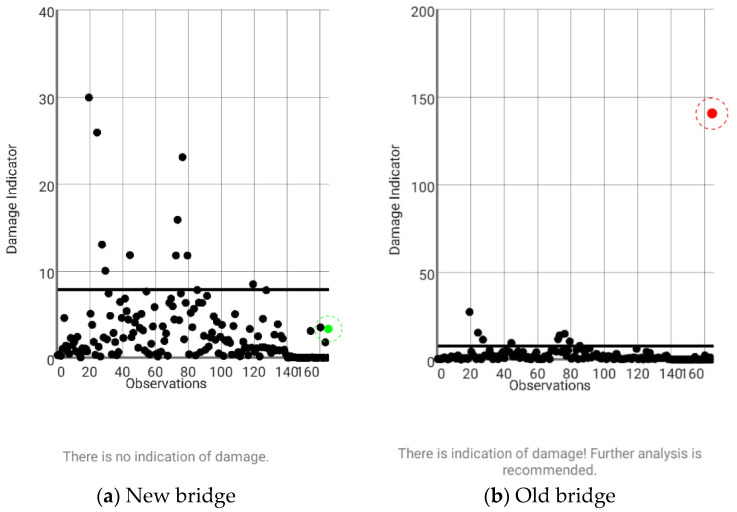
Damage detection at section 5 of new and old bridges using training data from the new bridge only.

**Table 1 sensors-22-08483-t001:** Values, mean, standard deviation, and 95% confidence interval of the steel beam’s natural frequencies estimated with three sensing systems.

Test No.	F1 (Hz)			F2 (Hz)			F3 (Hz)		
	Smartphone	393B12	333B40	Smartphone	393B12	333B40	Smartphone	393B12	333B40
1	2.119	2.120	2.120	8.616	8.657	8.657	19.774	19.788	19.788
2	2.210	2.120	2.120	8.702	8.657	8.657	18.370	19.788	18.375
3	2.187	2.120	2.120	8.601	8.657	8.657	18.367	18.375	18.375
4	2.107	2.120	2.120	8.708	8.834	8.834	18.258	18.375	18.375
5	2.180	2.120	2.120	8.866	8.834	8.834	18.314	19.788	19.788
6	2.083	2.120	2.120	8.780	8.834	8.657	18.304	18.375	18.375
7	2.199	2.315	2.315	8.798	8.796	8.796	18.475	18.519	18.519
8	2.237	2.222	2.222	8.852	8.778	8.778	18.482	18.444	18.444
9	2.107	2.120	2.120	8.708	8.834	8.834	18.258	18.198	18.198
10	2.202	2.120	2.120	8.679	8.657	8.657	18.264	18.198	18.198
11	2.190	2.252	2.252	8.759	8.709	8.709	18.370	18.468	18.468
12	2.257	2.252	2.252	8.789	8.859	8.859	18.527	18.468	18.468
13	2.243	2.252	2.252	8.707	8.709	8.709	18.470	18.468	18.468
14	2.296	2.252	2.252	8.801	8.859	8.859	18.495	18.468	18.468
15	2.255	2.120	2.120	8.753	8.657	8.834	18.435	18.375	18.375
16	2.228	2.120	2.120	8.635	8.657	8.657	18.384	18.375	18.375
17	2.242	2.219	2.219	8.857	8.747	8.747	18.498	18.407	18.407
18	2.290	2.297	2.297	8.779	8.834	8.834	18.448	18.375	18.551
19	2.297	2.297	2.297	8.784	8.834	8.834	18.514	18.551	18.551
20	2.344	2.297	2.297	8.854	8.834	8.834	18.490	18.551	18.551
21	2.179	2.252	2.252	8.718	8.709	8.709	18.462	18.468	18.468
22	2.279	2.297	2.297	8.847	8.834	8.834	18.499	18.551	18.375
23	2.304	2.297	2.297	8.808	8.834	8.834	18.428	18.551	18.551
24	2.309	2.222	2.222	8.835	8.889	8.889	18.474	18.444	18.444
25	2.326	2.297	2.297	8.915	8.834	8.834	18.605	18.551	18.551
26	2.213	2.219	2.219	8.853	8.747	8.747	18.511	18.407	18.407
27	2.332	2.297	2.297	8.808	8.834	8.834	18.523	18.551	18.551
28	2.244	2.297	2.297	8.728	8.657	8.657	18.454	18.375	18.375
29	2.255	2.297	2.297	8.753	8.657	8.657	18.568	18.551	18.375
30	2.232	2.252	2.252	8.705	8.709	8.709	19.754	19.820	19.820
Mean	2.232	2.219	2.219	8.767	8.765	8.765	18.526	18.621	18.568
SD	0.067	0.075	0.075	0.077	0.080	0.080	0.343	0.470	0.420
CI’s LL	2.207	2.193	2.193	8.739	8.736	8.736	18.403	18.453	18.418
CI’s UL	2.256	2.246	2.246	8.794	8.793	8.793	18.648	18.789	18.718

**Table 2 sensors-22-08483-t002:** Damage detection according to two algorithms.

Condition	Mahalanobis Squared Distance		Gaussian Mixture Model	
	Correct	Incorrect	Correct	Incorrect
Undamaged	96	4	96	4
25 g at mid-span	0	5	0	5
25 g at 1/4 of the span	0	5	0	5
50 g at mid-span	4	1	4	1
50 g at 1/4 of the span	2	3	2	3
100 g at mid-span	5	0	5	0
100 g at 1/4 of the span	5	0	5	0

**Table 3 sensors-22-08483-t003:** Comparison of the modal properties using the conventional system (GeoSIG).

Mode	Old Bridge (15 October 2021)	New Bridge (11 October 2021)
1	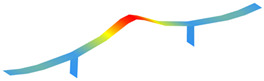	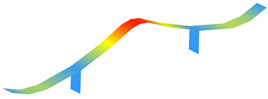
	1.471 Hz	1.025 Hz
2	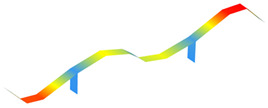	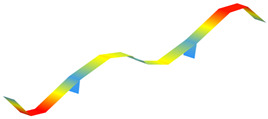
	2.620 Hz	1.731 Hz
3	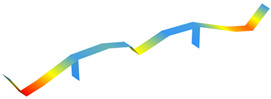	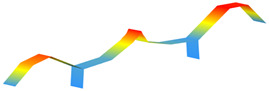
	2.832 Hz	2.100 Hz
4	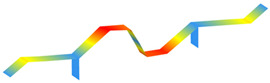	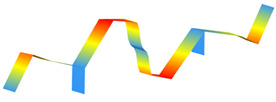
	3.816 Hz	2.905 Hz
5	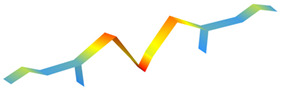	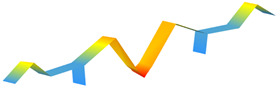
	5.845 Hz	4.582 Hz

**Table 4 sensors-22-08483-t004:** Comparison of the natural frequencies.

Mode	Old Bridge (15 October 2021)	New Bridge (11 October 2021)
1	1.471	1.025 (−30.3%)
2	2.620	1.731 (−33.9%)
3	2.832	2.100 (−25.8%)
4	3.816	2.905 (−23.9%)
5	5.845	4.582 (−21.6%)

**Table 5 sensors-22-08483-t005:** Estimation of the natural frequencies in section 2 of the new bridge over one hour (GeoSIG).

Starting Date and Hour	F2 (Hz)	F3 (Hz)	F4 (Hz)
11 October 21 16:20	1.725 (−0.37%)	2.094 (1.08%)	2.909 (0.28%)
11 October 21 16:40	1.731 (−0.02%)	2.066 (−0.27%)	2.891 (−0.35%)
11 October 21 17:00	1.738 (0.39%)	2.055 (−0.81%)	2.903 (0.07%)
**Mean**	**1.731**	**2.072**	**2.901**

**Table 6 sensors-22-08483-t006:** Estimation of the natural frequencies at section 2 (new bridge) using App4SHM and comparison with the means of the reference system (GeoSIG) from Table 5 (in bold).

Starting Date and Hour	F2 (Hz)	F3 (Hz)	F4 (Hz)
11 October 21 16:12	1.717 (−0.83%)	2.039 (−1.58%)	2.897 (−0.14%)
11 October 21 16:16	1.757 (1.48%)	2.027 (−2.16%)	2.838 (−2.17%)
11 October 21 16:22	1.772 (1.35%)	2.025 (−2.53%)	2.885 (−0.55%)
11 October 21 16:33	1.738 (0.39%)	2.070 (−0.08%)	2.897 (−0.14%)
11 October 21 16:38	1.757 (1.48%)	2.030 (−2.01%)	2.907 (0.21%)
11 October 21 16:46	1.767 (2.06)	2.061 (−0.52%)	2.886 (−0.52%)
Mean	1.751 **(1.16%)**	2.042 **(−1.43%)**	2.914 **(−0.55%)**

**Table 7 sensors-22-08483-t007:** Comparison between GeoSIG and App4SHM sensing systems throughout an ambient vibration test (old bridge).

Sensing System	F1 (Hz)	F3 (Hz)	F5 (Hz)
GeoSIG	1.471	2.832	5.845
App4SHM	1.507 (2.42%)	2.848 (0.55%)	5.842 (−0.05%)

## Data Availability

Not applicable.

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
