# Peer review of "Smartphone Application for Structural Health Monitoring of Bridges"

_sensors, 2022, doi:10.3390/s22218483_

Round 1

Reviewer 1 Report

This paper presents a smartphone application called App4SHM, as a customized SHM process for damage detection. It is interesting and practical significant. However, the reviewer has the following concerns. The authors should address these concerns before the manuscript being recommended to be published.

1. What is the required mobile phone hardware configuration (such as the parameters of the built-in accelerometer)? Can all commonly used mobile phones be used for damage detection?

2. In the Damage detection stage, the author applies a Mahalanobis-based algorithm to detect abnormal observations by measuring the distance between the previously recorded frequencies and the current observation. This approach is too simple to accurately determine whether the structure is damaged. The feasibility of this method needs to be explained in more detail.

3. In the field test, there are many influencing factors, such as noise, temperature, vehicles, etc. How can the author eliminate these influencing factors and identify accurate natural frequency values for damage detection algorithm.

Author Response

The authors would like to express their gratitude for the comments of the reviewers. The comments are answered below. All references in this text refer to the revised version of the article.

Reviewer 1 

Question: What is the required mobile phone hardware configuration (such as the parameters of the built-in accelerometer)? Can all commonly used mobile phones be used for damage detection?

Answer: The application runs on any mobile phone with Android 10 or superior. All smartphones include an accelerometer that is adequate for the data acquisition stage, without specific configuration.

Changes: We included this information on lines 180-1.

Q: In the Damage detection stage, the author applies a Mahalanobis-based algorithm to detect abnormal observations by measuring the distance between the previously recorded frequencies and the current observation. This approach is too simple to accurately determine whether the structure is damaged. The feasibility of this method needs to be explained in more detail.

A: The Mahalanobis Squared Distance is a standard damage detection procedure, successfully used in many SHM publications (e.g. [25,26]). We agree, however, that the choice of the method is not well explained in the first version of the paper and that the demonstration of its performance using the twin Itacaiúnas bridges is not sufficiently convincing. We thank the reviewer for the suggestion.

C: Therefore, a new section (3.4) was added, testing the damage detection performance of the application (using the Mahalanobis Squared Distance) and comparing it with that of a Gaussian Mixture Model, which is another mainstream damage detection technique. The performances are similar. Moreover, a new paragraph was added to Section 2.2, where the choice of the Mahalanobis Squared Distance algorithm is justified (lines 284-90).

Q: In the field test, there are many influencing factors, such as noise, temperature, vehicles, etc. How can the author eliminate these influencing factors and identify accurate natural frequency values for damage detection algorithm.

A: Those influencing factors are taken into account in different ways. First, the app measures the bridge response under ambient vibration tests, which relies on the traffic caused by vehicles and people to excite the structure. Second, undesired influencing factors like noise and temperature are removed or minimized through digital transformations performed with the Welch algorithm (by overlaping segments and hamming windows) and through the normalization process performed by the machine learning algorithm based on the Mahalanobis Squared Distance.

C: No changes were made to the manuscript.

Reviewer 2 Report

How was the sampling frequency set? Justify.

Include a visible experimental setup showcasing complete instrumentation.

Justify the orientation of an accelerometer.

Why only the vibration parameter was considered when you call it structural health monitoring?

Add a comparison of the proposed and other available methods in tabular form w.r.t. different factors.

What’s the scope of a MEMS-based accelerometer integrated with an open source controller such as Arduino etc. for measurement of vibrations? Refer to the articles, ‘Novel Machine Health Monitoring System,’ ‘Application of Machine Learning for Tool Condition Monitoring in Turning,’ and ‘Tyre Pressure Supervision of Two Wheeler Using Machine Learning' for an idea of developing in-house low-cost DAQ. You may suggest this in future scope.

In continuation, try connecting your work to the open-design movement reviewed in the paper ‘Overview of contemporary systems driven by open-design movement’. This would open new opportunities for authors and readers as well.

What’s the scope of IoT in this structural health monitoring using smartphones?

What’s the scope of AI-based methods in this structural health monitoring using smartphones?

The grammar is improvable in a few places. There must be thorough proofreading of the paper.

Author Response

The authors would like to express their gratitude for the comments of the reviewers. The comments are answered below. All references in this text refer to the revised version of the article.

 Reviewer 2 

Question: How was the sampling frequency set? Justify.

Answer: The sampling frequency of the accelerometers in most smartphones is of 50 Hz. However, smartphones have imperfect sampling frequencies, so an interpolation algorithm is implemented to correct deviations from the defined sampling frequency.

Changes: This is mentioned in the paper, see lines 245-248.

Q: Include a visible experimental setup showcasing complete instrumentation.

A: We agree with the reviewer.

C: As the accelerometers from PCB Piezotronics are glued to the underside of the steel plate, we kept Figure 6, where they are presented, and added the new Figure 12, where the other elements from the setup are presented.

Q: Justify the orientation of an accelerometer.

A: Since the smartphone is simply laid on the surface of the bridge (or of the steel beam), the choice of the measurement axis is designed to measure the acceleration along its main direction of vibration. 

C: We mentioned this explanation in the manuscript on lines 248-9.

Q: Why only the vibration parameter was considered when you call it structural health monitoring?

A: SHM systems involve many tools of diagnostic, from visual inspection to bathymetric measurements, and from climatic monitoring to vibration measurements. In this sense, App4SHM is not meant to be used unaccompanied, but integrated into a Bridge Management System, along with complementary sensors and non-destructive testing.

C: A phrase acknowledging the need for complementary measurement and diagnostic tools was added to the conclusions (lines 635-7).

Q: Add a comparison of the proposed and other available methods in tabular form w.r.t. different factors

A: We agree that the choice of the method is not well explained in the first version of the paper and that the demonstration of its performance using the twin Itacaiúnas bridges is not sufficiently convincing. We thank the reviewer for the suggestion.

C: A new section (3.4) was added, testing the damage detection performance of the application and comparing it with that of a Gaussian Mixture Model, which is another mainstream damage detection technique. The performances are similar. Moreover, a new paragraph was added to Section 2.2, where the choice of the Mahalanobis Squared Distance algorithm is justified.

Q: What’s the scope of a MEMS-based accelerometer integrated with an open source controller such as Arduino etc. for measurement of vibrations? Refer to the articles […]

A: We thank the reviewer for the suggestions. In fact, App4SHM does fit into a larger effort to transpose SHM technology from research to practice by developing innovative, low-cost data acquisition systems along with supporting software.

C: Two papers fitting the above description were cited, see lines 161-3.

Q: In continuation, try connecting your work to the open-design movement reviewed in the paper ‘Overview of contemporary systems driven by open-design movement’.

A: We thank the reviewer for the suggestion. The open-design movement is indeed an altruistic approach to the development of new products. We are in the final stages of the preparation of a paper on a novel, modified Rowe cell device for dynamic measurements on geomaterials. Its whole design is openly presented, so it is probably aligned with the open-design concept. App4SHM, however, does not seem to fit such scope. On the one hand, because it is a software, so the open-source concept seems to be a better fit than open-design. On the other hand, because it is a tool for companies aiming at making a profit, so its final version will probably not be given away for free.

C: No changes were made to the manuscript.

Q: What’s the scope of IoT in this structural health monitoring using smartphones?

A: As mentioned above, App4SHM is not meant to be used solo, but integrated into a Bridge Management System with complementary sensors and assessments from expert bridge inspectors. Such systems are typically complex software, connected to the network of sensors (IoT) and accepting input from the bridge inspectors. App4SHM is conceived to be easy to integrate in and add value to such software.

C: No changes were made to the manuscript.

Q: What’s the scope of AI-based methods in this structural health monitoring using smartphones?

A: In its multiple forms, AI stands at the base of Machine Learning Algorithms for damage detection. As typical to Civil Engineering applications, damage detection is performed in an unsupervised mode, where the distinction damaged-undamaged is pursued, provided a consistent database of readings under undamaged conditions is available. 

C: No changes were made to the manuscript.

Q: The grammar is improvable in a few places. There must be thorough proofreading of the paper

A: The paper was proofread.

C: Grammar and expression errors were corrected.

Round 2

Reviewer 2 Report

The authors have revised the manuscript as per the comments. I'm impressed. The quality has been improved tremendously. All the best. I recommend its acceptance.